# Relative Effects of Brachytherapy and Beam Radiation for DCIS on Subsequent Invasive Events

**DOI:** 10.3390/biomedicines13112823

**Published:** 2025-11-19

**Authors:** Thomas J. O’Keefe, Nicolas D. Prionas, Anne M. Wallace

**Affiliations:** 1Department of Surgery, University of California San Diego, San Diego, CA 92037, USA; 2Department of Radiation Oncology, University of California San Francisco, San Francisco, CA 94158, USA

**Keywords:** DCIS, breast cancer outcomes, brachytherapy, partial-breast irradiation, whole-breast irradiation, breast cancer, breast cancer recurrence, breast cancer outcomes

## Abstract

**Background/Objectives:** Patients with a diagnosis of ductal carcinoma in situ (DCIS) were poorly represented in the four trials that established the efficacy of partial relative to whole-breast irradiation. In contrast to invasive cancers, patients with DCIS are equally likely to have a subsequent ipsilateral invasive event in a different site of the breast from their initial DCIS lesion as they are at the same site. We aim to compare the efficacy of a type of partial-breast irradiation, brachytherapy, to external-beam radiation in the reduction of subsequent invasive cancers. **Methods:** Women diagnosed with a first breast cancer of unilateral DCIS treated with breast-conserving surgery without endocrine therapy were identified in SEER. Matching was performed 1:2 from patients receiving brachytherapy to patients receiving external-beam radiation or no radiation. External-beam radiation was assumed to be whole-breast radiation for the majority of patients in this cohort diagnosed from 2007 to 2011. Competing risks methods were used to estimate the cumulative incidence of invasive breast events at 10 years and subdistribution hazard ratios (sHRs) in adjusted models with time-varying treatment coefficients were calculated. **Results:** Among the 1392 matched patients who received brachytherapy or no radiation, the 10-year cumulative incidence of ipsilateral invasive cancer was 5.5% without radiation and 5.7% with brachytherapy (*p* = 0.92). Brachytherapy was associated with reduced risk of ipsilateral invasive events in the first 3 years (sHR = 0.19, *p* = 0.03) and non-significantly increased risk after 3 years (sHR = 1.66, *p* = 0.07). Among the 1392 matched patients who received brachytherapy or external-beam radiation, the 10-year cumulative incidence of ipsilateral invasive cancer was 5.7% with brachytherapy and 3.1% with external-beam radiation. In multivariate regression, brachytherapy was associated with unchanged risk in the first 3 years but increased risk after 3 years (sHR = 2.20, *p* = 0.009). **Conclusions:** Our results suggest that brachytherapy may be associated with a higher invasive recurrence risk for patients with DCIS treated without endocrine therapy, as it did not prevent more invasive ipsilateral events than no radiation, and provided reduced risk reduction in the ipsilateral breast relative to external-beam radiation. Further work is needed to identify if this is specific to brachytherapy or applies to all forms of partial-breast irradiation.

## 1. Introduction

Ductal carcinoma in situ (DCIS) is a heterogeneous disease process that represents both a global high-risk marker for subsequent invasive disease and a preinvasive lesion with intrinsic malignant potential. The early randomized trials for DCIS assessed the effects of adjuvant conventional whole-breast irradiation (WBI) and consistently found that such treatment reduced the risk of subsequent in situ and invasive ipsilateral events [1,2,3,4,5]. Four subsequent trials enrolled patients diagnosed with either DCIS or early invasive breast cancer to receive partial-breast irradiation (PBI) or WBI. While the absolute differences in ipsilateral breast tumor recurrence rates between PBI and WBI were small in all four trials, two found statistically significant differences between the groups and the other two did not [6,7,8,9]. The two trials that did not find a difference exclusively utilized external-beam PBI [6,7], whereas the two that found a difference either allowed [9] or exclusively utilized [8] brachytherapy for PBI. Post hoc analysis from NSABP B39 suggested that patients with a lesion size measuring 11–20 mm benefited more from WBI, while patients who were perimenopausal or had hormone receptor positive disease may have benefited more from WBI. These trials helped inform modern ASTRO guidelines, which offer a strong recommendation for PBI in patients with DCIS who are 40 years or older, have low- or intermediate-grade disease, and have a lesion measuring 2 cm or less.

Notably, while patients with DCIS and early invasive disease were grouped together for this study, important differences exist between the recurrence patterns of these two disease processes. For patients with invasive disease, ipsilateral breast tumor recurrences occur in or near the same quadrant of the breast as the initial cancer roughly 95% of the time [10]. In contrast, for patients with DCIS, only 52% of invasive recurrences occur in or near the same quadrant, while 80% of in situ recurrences occur in or near the same quadrant [11]. Since only invasive subsequent events and not in situ recurrences have been associated with an increased risk of breast cancer mortality for DCIS patients [3,12], it is possible that PBI could preferentially reduce the less important in situ recurrence rate at the expense of decreased efficacy against invasive subsequent events. Since the previously mentioned trials did not differentiate in situ from invasive recurrences, and DCIS patients were relatively poorly represented among them, ranging from a low of 5% of patients in GEC-ESTRO to a high of 24% of patients in NSABP B39/RTOG 0413 [8,9], it is unclear from the existing trials whether PBI has a differential effect on invasive and in situ ipsilateral recurrences relative to WBI.

Here we aim to use data from a large national registry to identify the effects of brachytherapy on ipsilateral invasive breast cancer recurrences in patients with DCIS compared to external-beam radiation or no radiation. In contrast to the trials on PBI, we focus exclusively on invasive events, the most important prognostic recurrence type for patients with DCIS. We restricted our analysis to patients who did not receive endocrine therapy to isolate the effects of radiotherapy, so our findings best generalize to DCIS treated without endocrine therapy.

## 2. Materials and Methods

### 2.1. Ethical Considerations

The UCSD Institutional Review Board deferred the need for approval of this study due to its use of publicly available de-identified national registry data. The study was conducted in accordance with the US Common Rule.

### 2.2. Cohort Creation

Data from the Surveillance, Epidemiology and End Results (SEER) 17 registry grouping (November 2023 submission) were used. Data were extracted in April 2024. Female patients whose first cancer event was unilateral DCIS treated with breast-conserving surgery with follow-up greater than 6 months, who were 75 years or younger, had a tumor measuring 3 cm or smaller, and had a known cause of death, race, year of diagnosis, lesion laterality, surgery type, grade, estrogen receptor (ER) status, and progesterone receptor (PR) status were included. Patients were excluded if they had known HER2 status, as HER2 testing is not the standard of care for patients with DCIS and may influence the risk of ipsilateral invasive recurrence [13,14]. They were also excluded if they received endocrine therapy to avoid confounding [4,12]. The age cutoff was selected to avoid undue influence of the competing risk of non-breast cancer deaths for elder patients, and the size cutoff was selected to minimize the effect of outliers, given the single dimension provided for size. Patients had to be diagnosed at least ten years prior to the year of final follow-up for data in the SEER iteration. Patients who received chemotherapy or neoadjuvant radiation therapy were excluded, as this is not the standard of care for DCIS. Patients who received neoadjuvant endocrine therapy were excluded as this is not the standard of care. The ipsilateral and contralateral invasive events and times to these events were collected, and a composite variable of “any invasive event” was developed to represent the first of any invasive diagnoses that the patient received, regardless of laterality, as well as breast cancer mortality in the absence of an in-breast invasive event. Breast cancer mortality was also considered as a separate outcome. Time to last follow-up or time to non-breast cancer death were also collected.

### 2.3. Matching

Propensity scores using a binomial model with the variables of tumor size, patient race, patient age, year of diagnosis, and exact matching on tumor grade and hormone receptor status were calculated, and 1:2 pair matching without replacement was performed between groups using the optimal method, with rank-based Mahalanobis matching within propensity score calipers of size 0.20 standard deviations. The 1:2 ratio was selected due to the relatively small numbers of patients who received brachytherapy relative to those not receiving radiotherapy and those receiving external-beam radiation. Cumulative incidences of the first of any invasive events, including breast cancer mortality, were estimated using Fine and Gray competing risks methods [15]. Non-breast cancer death was treated as a competing risk for invasive events. Ipsilateral invasive same-quadrant and different-quadrant events were assessed, with any death treated as the only competing event. Quadrants were derived from ICD; primary site codes were used to determine this. For recurrence quadrant analysis, only sites with a defined quadrant (upper/lower and inner/outer) were included for this portion of the analysis, and patients with borderline and “not otherwise specified” sites were not considered. Multivariate models used the following variables: patient age (55 years or younger vs. older than 55 years), hormone receptor status (ER or PR positive vs. ER and PR negative), tumor size (10 mm or less vs. greater than 10 mm), tumor grade (low or intermediate vs. high or undifferentiated), patient race (white or non-white, non-Black vs. Black), and treatment (which was handled with a time-dependent coefficient with a cutoff at 3 years—this time was selected by graphical inspection), and subdistribution hazard ratios (sHRs) were calculated. Age and size cut points were selected based on distributional considerations.

Statistical significance was declared for *p* less than 0.05. All statistical analyses were performed in R (4.4.1, R Foundation for Statistical Computing, Vienna, Austria) using RStudio (2024.09.0+375) and the packages “optmatch” (0.10.8), “tidyverse” (2.0.0), “cmprsk” (2.2–12), and “survival” (3.8–3) [16,17,18].

## 3. Results

### 3.1. Unmatched Cohort

A total of 7960 patients meeting the criteria were identified. The median age of this group was 57 (interquartile range [IQR] 50–65), and the median tumor size was 8 mm (IQR 5–15). Patients were treated between 2007–2011. Among them, 464 (5.8%) received brachytherapy, 2509 (31.5%) did not receive adjuvant radiation, and 4987 (62.7%) received external-beam radiation. A total of 6278 (78.9%) patients were white, 724 (9.1%) patients were Black, and 958 (12.0%) patients were neither white nor Black. A total of 994 (12.5%) patients had low-grade disease, 3453 (43.4%) patients had intermediate-grade disease, 2661 (33.4%) patients had high-grade disease, and 852 (10.7%) patients had undifferentiated-grade disease. A total of 6500 (81.7%) patients had estrogen or progesterone receptor-positive disease and 1460 (18.3%) had estrogen and progesterone receptor-negative disease. None of the patients received endocrine therapy, as such patients were excluded to avoid confounding. A total of 279 patients (3.5%) had an ipsilateral invasive event in the 10-year period, 261 (3.3%) had a contralateral invasive event, 66 (0.8%) died of breast cancer, and 472 (5.9%) died of non-breast cancer causes.

The 10-year cumulative incidence of an ipsilateral invasive subsequent event in the full cohort was 4.8% for patients who did not receive adjuvant radiation, 5.7% for patients who received brachytherapy, and 2.7% for patients who received external-beam radiation. The 10-year cumulative incidence of a contralateral invasive subsequent event was 2.6% among patients who did not receive adjuvant radiation, 3.9% among patients who received brachytherapy, and 3.6% among patients who received external-beam radiation. The 10-year cumulative incidence of any subsequent invasive event was 7.5% among patients who underwent breast-conserving surgery without radiation, 10.2% among patients who received brachytherapy, and 7.8% among patients who received external-beam radiation.

### 3.2. Matched Brachytherapy vs. Omission of Radiation

A total of 1392 patients meeting criteria were identified (Table 1). The median age of this group was 60 (IQR 53–66.25), and the median tumor size was 8 mm (IQR 4–14). Among them, 464 (33.3%) received brachytherapy and 928 (66.7%) did not receive adjuvant radiation. A total of 1153 (82.8%) patients were white, 136 (9.8%) patients were Black, and 103 (7.9%) patients were neither white nor Black. A total of 138 (9.9%) patients had low-grade disease, 624 (44.8%) patients had intermediate-grade disease, 483 (34.7%) patients had high-grade disease, and 147 (10.6%) patients had undifferentiated-grade disease. A total of 1104 (79.3%) patients had estrogen or progesterone receptor-positive disease and 288 (20.7%) had estrogen and progesterone receptor-negative disease. None of the patients received endocrine therapy, as such patients were excluded to avoid confounding. A total of 76 patients (5.5%) had an ipsilateral invasive event in the 10-year period, 43 (3.1%) had a contralateral invasive event, 18 (1.3%) died of breast cancer, and 106 (7.6%) died of non-breast cancer causes.

The 10-year cumulative incidence (Figure 1) of any invasive subsequent event was 8.3% for patients who received no adjuvant radiation and 10.2% for patients who received brachytherapy (*p* = 0.27). The 10-year cumulative incidence of an ipsilateral invasive subsequent event was 5.5% for patients who did not receive adjuvant radiation and 5.7% for patients who received brachytherapy (*p* = 0.92). The 10-year cumulative incidence of a contralateral invasive subsequent event was 2.7% among patients who did not receive adjuvant radiation and 3.9% among patients who received brachytherapy (*p* = 0.24). Breast cancer mortality at 10 years was 1.5% among patients who received brachytherapy and 1.2% among patients who did not receive adjuvant radiation. In multivariate competing risks models (Table 2), brachytherapy was associated with a reduced risk of ipsilateral invasive subsequent events in the first 3 years (sHR = 0.19, *p* = 0.02) and was non-significantly associated with increased risk of subsequent ipsilateral invasive events after 3 years (sHR = 1.65, *p* = 0.07). Brachytherapy was also associated with no change in contralateral invasive events in the first 3 years (sHR = 0.29, *p* = 0.24) and non-significantly associated with increased risk of a contralateral invasive event after 3 years (sHR = 1.91, *p* = 0.06). For any invasive subsequent events, brachytherapy was associated with decreased risk in the first 3 years (sHR = 0.28, *p* = 0.02) followed by a significantly increased risk of invasive events after 3 years (sHR = 1.81, *p* = 0.005).

### 3.3. Matched Brachytherapy vs. Beam Radiation

A total of 1392 patients meeting criteria were identified (Table 1). The median age of this group was 60 (IQR 53–67), and the median tumor size was 8 mm (IQR 5–14). Among them, 464 (33.3%) received brachytherapy and 928 (66.7%) received external-beam radiation. A total of 1157 (83.1%) patients were white, 136 (9.8%) patients were Black, and 103 (7.1%) patients were neither white nor Black. A total of 138 (9.9%) patients had low-grade disease, 624 (44.8%) patients had intermediate-grade disease, 483 (34.7%) patients had high-grade disease, and 147 (10.6%) patients had undifferentiated-grade disease. A total of 1104 (79.3%) patients had estrogen or progesterone receptor-positive disease and 288 (20.7%) had estrogen and progesterone receptor-negative disease. None of the patients received endocrine therapy, as such patients were excluded to avoid confounding. A total of 54 patients (3.4%) had an ipsilateral invasive event in the 10-year period, 49 (3.5%) had a contralateral invasive event, 14 (1.0%) died of breast cancer, and 79 (5.7%) died of non-breast cancer causes.

The 10-year cumulative incidence (Figure 2) of any invasive subsequent event was 10.2% for patients who received brachytherapy and 7.0% for patients who received external-beam radiation (*p* = 0.04). The 10-year cumulative incidence of an ipsilateral invasive subsequent event was 5.7% for patients who received brachytherapy and 3.1% for patients who received external-beam radiation (*p* = 0.02). The 10-year cumulative incidence of a contralateral subsequent invasive event was 3.9% for patients who received brachytherapy and 3.4% for patients who received external-beam radiation (*p* = 0.63). The 10-year cumulative incidence of breast cancer mortality was 1.5% for patients who received brachytherapy and 0.8% for patients who received external-beam radiation (*p* = 0.19). In multivariate competing risks models (Table 3), brachytherapy was associated with no change in risk during the first 3 years (sHR = 0.65, *p* = 0.59) and an increased risk after 3 years (sHR = 2.19, *p* = 0.009). Brachytherapy was associated with unchanged risk of contralateral subsequent invasive events in both the first 3 years (sHR = 0.33, *p* = 0.31) and after 3 years (sHR = 1.36, *p* = 0.34). For any invasive subsequent event, brachytherapy was associated with no change in risk during the first 3 years (sHR = 0.61, *p* = 0.38) and an increased risk after 3 years (sHR = 1.72, *p* = 0.009).

### 3.4. Site of Ipsilateral Breast Invasive Subsequent Event Relative to Initial DCIS Location

We performed an exploratory analysis regarding whether patients with well-defined index in situ and ipsilateral invasive recurrence were located in the same or different quadrants. Patients whose initial DCIS and subsequent ipsilateral invasive cancer were both diagnosed in a specified quadrant of the breast (upper-outer, upper-inner, lower-outer, lower-inner) were compared on whether the initial and subsequent lesion locations were the same or different. Among patients meeting these criteria, 11 of 20 (55.0%) patients who did not receive adjuvant radiation had lesions in the same quadrant, 4 of 14 (28.6%) patients who received brachytherapy had lesions in the same quadrant, and 4 of 10 (40.0%) patients who received external-beam radiation had lesions in the same quadrant.

Among the 20 patients who did not receive adjuvant radiation and met these criteria, in the first 3 years, 8 of 11 (72.7%) ipsilateral invasive events occurred at the same site as the initial DCIS lesion, whereas after 3 years only 3 of the 9 (33.3%) ipsilateral invasive subsequent events occurred in the same quadrant as the initial DCIS lesion.

## 4. Discussion

Here we demonstrate in a large cohort of patients with DCIS that (i) relative to PBI or whole-breast external-beam radiation, brachytherapy was associated with increased risk of ipsilateral subsequent invasive events, (ii) relative to omission of radiation therapy, brachytherapy was not associated with reduced risk of ipsilateral invasive events, but rather shifted the timing of ipsilateral invasive subsequent events from earlier to later in the 10-year period and the location from the same quadrant of the breast as the DCIS to different quadrants, and (iii) among patients who did not receive radiation, early subsequent ipsilateral invasive events were more commonly at the same site, whereas late ipsilateral invasive events were more commonly at different sites. When considering the outcome of an invasive in-breast subsequent event in either breast or breast cancer mortality, brachytherapy was associated with a significantly increased risk relative to patients who received external-beam radiation and a non-significantly increased risk relative to patients who did not receive adjuvant radiation. This suggests that brachytherapy may be associated with a greater risk of invasive events, which are more clinically important than in situ subsequent events. Of note, due to the manner in which data is coded in SEER, we were unable to differentiate patients who received external-beam PBI from those who received external-beam WBI, so it is possible that this finding relates generally to PBI rather than to brachytherapy specifically, but we are unable to assess for this.

While patients diagnosed with DCIS were included in the four previously referenced randomized trials that assessed the efficacy of PBI relative to WBI, there are several important limitations with respect to the generalizability of these findings to DCIS [6,7,8,9]. First, patients with DCIS were poorly represented in these trials, ranging from 5% to 24% of patients enrolled in each, and only one of these four trials performed a subgroup analysis of patients with DCIS [8,9]. Second, while the overwhelming majority of patients diagnosed with invasive cancer receive systemic therapy, many patients diagnosed with DCIS do not, representing an important confounder. Third, the outcome of interest in these trials was ipsilateral breast tumor recurrences, but invasive subsequent events, regardless of laterality, are the most important events for patients with an initial diagnosis of DCIS, as they represent an upstaging of disease, typically necessitate systemic therapy, and are associated with a significantly increased risk of breast cancer mortality [3,12]. Even within the ipsilateral breast, subsequent invasive events occur in different quadrants of the breast from the initial DCIS lesion in roughly half of cases, relative to about 80% of in situ subsequent events [11]. These observations were the primary motivators for this study.

If we simplistically assume that the majority of patients in the external-beam radiation group received WBI rather than PBI, a not-entirely unreasonable assumption given that ASTRO guidelines did not begin to recommend PBI for DCIS until 2017 [19], five years after any of the patients in our study were treated, then our findings with respect to brachytherapy vs. external-beam radiation could be explained by the pattern of site of recurrence noted among patients treated without radiation therapy. That is, since patients treated without radiation are more likely to have invasive ipsilateral events at the same site as their initial DCIS in the first few years and are more likely to develop invasive events at other sites in later years, the time-dependent effect of brachytherapy could be explained by the preferential effect on same-site invasive events, since these are more likely to occur in the short-term, whereas differences in different-quadrant invasive subsequent events, which might be reduced by WBI given its whole-breast coverage, would be less likely to manifest until years after treatment, as observed here.

We also assessed the relative efficacy of omission of radiation and brachytherapy. This has not been assessed in randomized trials, as part of the rationale for the inclusion of DCIS in trials of PBI was the low rate of ipsilateral breast tumor recurrences in RTOG 9804, in which patients with low-risk DCIS (2.5 cm or less, margins 3 mm or greater, screen-detected, low- or intermediate-grade) were randomized to WBI (most received conventional, 10% received hypofractionated) or no adjuvant radiation. That trial, however, like the four trials of PBI for early invasive breast cancer and DCIS, was confounded by systemic therapy, with more than half of patients receiving tamoxifen. We found that brachytherapy was associated with a reduction in the risk of ipsilateral invasive events in the first 3 years, but was non-significantly associated with an increased risk of ipsilateral invasive events after 3 years, such that the cumulative incidence of ipsilateral invasive events was nearly identical between the two groups by 10 years, with an absolute difference of only 0.2%. The net effect on the outcome of any invasive event in either breast or breast cancer mortality was a significantly reduced with brachytherapy in the first 3 years, followed by a significantly increased risk after 3 years.

We performed an exploratory analysis on the locations of initial DCIS lesions and subsequent invasive lesions when they were listed as being located in a single quadrant and not on the line between quadrants or covering multiple quadrants. Invasive subsequent events were more likely to occur in the same quadrant as the initial DCIS lesion in patients who did not receive adjuvant radiation, and they were more likely to have invasive subsequent events in different quadrants in patients who received brachytherapy. If this association were representative of the patients whose initial and subsequent lesion locations could not be as easily compared, it would suggest that brachytherapy is effective at preventing invasive subsequent events in the vicinity of the initial lesion, but it may also increase the risk of invasive events in the ipsilateral breast farther away from the site of the initial lesion. This could also relate to our finding of temporal dependence on the location of subsequent invasive events for patients who did not receive radiation: The majority were at the same site when they occurred in the first 3 years but different sites after 3 years. If brachytherapy increased the risk of these different-quadrant invasive events, they may similarly manifest at a later time, which would explain the early decrease in ipsilateral invasive events and late increase.

We noted a borderline increased risk of contralateral invasive events among patients who underwent brachytherapy relative to those who did not receive radiation therapy in the 3–10 year time frame. There are multiple possible explanations for this observation. This could suggest differences in baseline global risk for breast cancer, and suggestive of confounding by unobserved variables such as family history. However, it should be noted that of the three trials of radiotherapy for DCIS that reported on cumulative contralateral invasive cancer risks, SweDCIS identified a borderline significantly elevated risk of contralateral invasive cancers, EORTC 10853 found a significantly elevated risk of contralateral invasive cancers, and only NSABP B17 did not identify an increased risk of contralateral cancers. This suggests that the borderline elevated increased risk may relate to radiation itself, rather than to PBI specifically.

Taken altogether, our findings suggest that brachytherapy may be associated with a higher risk of invasive events than external-beam radiation. Importantly, a motivating observation for this study was the observation that invasive ipsilateral events occur about as equally in the same location and different locations of the ipsilateral breast, but this does not apply to in situ recurrences, which occur more commonly at the same site as the initial DCIS. It is possible, then, that brachytherapy may have a more favorable profile for reducing in situ recurrences. However, it should be noted that most women will not have either an in situ or invasive ipsilateral event, independent of the use of radiation, and a woman who does not receive radiation for her initial DCIS lesion and develops an in situ ipsilateral recurrence still has the options of both PBI and WBI, regardless of the timing or site of her prior DCIS. Alternately, our results may suggest that the margin of brachytherapy may need to be increased for patients treated for DCIS relative to patients treated for early invasive breast cancer.

Our study’s greatest strength is the ability to compare a uniform cohort of patients in the absence of endocrine therapy, a known confounder that was present in all of the trials that have randomized patients with DCIS or early invasive cancers, to PBI or WBI. Our study has several limitations. As with all retrospective studies, we were unable to account for unobserved variables that may have confounded the results. There are several known confounders that were not available for our analysis: margin status, focality of disease/growth distribution, use of a radiation boost, family history or any known germline pathogenic mutations, and whether lesions were diagnosed by screening or by palpation. Bias may have been introduced by excluding known HER2 status, though only a minority of eligible patients (~5%) were excluded for this reason. SEER does not contain data regarding toxicity, cosmesis, patient-reported outcomes, or detailed dose or technique factors that can influence the choice between PBI and WBI. There have been changes in brachytherapy applicators over time, with differences among applicators in margin and conformality. Intraoperative radiotherapy was not included in this study, but it is a form of brachytherapy that is utilized at some institutions. A significant limitation was that while we were able to separate patients with brachytherapy from those receiving external-beam radiation, we could not separate patients receiving external-beam PBI from those receiving WBI, so our findings cannot be applied to PBI in general, only to patients who received brachytherapy. SEER also does not include information about tumor bed boosts or the specifics of the brachytherapy, such as whether it was interstitial or cavitary and whether a low- or high-dose rate was used. An additional limitation is that in our assessment of the location of the quadrant of the initial DCIS lesion and subsequent invasive event, we were only able to compare patients with defined lesion locations in the four quadrants, and could not account for lesions that were, for example, located on the lines between two quadrants, or at sites “not otherwise specified.” Similarly, it is possible that lesions could have been misclassified—the order of preference by SEER tumor registrars is listed as operative reports, pathology reports, then imaging such as mammogram or ultrasound, then physical exam. Furthermore, different modalities relied on for the initial and subsequent events would presumably increase the likelihood of them being located in different quadrants.

Prospective validation of our findings is indicated given these limitations. However, given the previously discussed gaps in knowledge from the existing clinical trials, our results suggest that for patients who will not receive endocrine therapy after breast-conserving surgery for DCIS, WBI may be preferable to PBI when the priority is the reduction of risk of invasive subsequent events. For example, while the magnitude of benefit in invasive risk reduction likely does not justify WBI over PBI for the majority of patients who presently meet ASTRO-specified criteria for PBI, patients with low clinicopathologic risk but high molecular risk may want to consider WBI, and similarly, patients who are on the younger end of the age criteria or who have a long life expectancy may similarly want to consider WBI given our finding of increasing risk with longer follow-up time.

## 5. Conclusions

In conclusion, we demonstrate here that for patients diagnosed with DCIS, brachytherapy is associated with reduction in risk of invasive cancers in the first 3 years after treatment, but an increase in the risk of invasive cancers after 3 years relative to patients who did not receive radiation. It was of comparable efficacy to external-beam radiation, assumed to predominantly be WBI, in the first 3 years after diagnosis, but presents an increased risk of ipsilateral invasive cancers after 3 years. For patients with a life expectancy greater than 10 years, our results suggest that brachytherapy may not confer as great of invasive risk reduction for patients treated with breast-conserving surgery for DCIS than external-beam radiation, as it does not change the long-term risk of invasive cancers relative to no radiation, nor does it reduce the risk of ipsilateral invasive cancers as effectively as external-beam radiation. Further work is needed to identify whether this applies to PBI as a whole or is specific to brachytherapy.

## Figures and Tables

**Figure 1 biomedicines-13-02823-f001:**
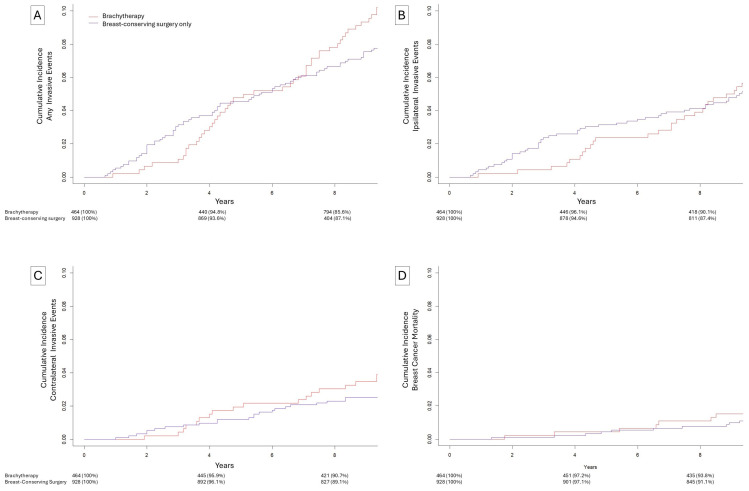
Cumulative incidence of events after breast-conserving surgery without radiation (blue) or brachytherapy (red). (**A**) Ipsilateral, contralateral, or breast cancer mortality subsequent events, (**B**) ipsilateral invasive subsequent events, (**C**) contralateral invasive subsequent events, (**D**) breast cancer specific mortality.

**Figure 2 biomedicines-13-02823-f002:**
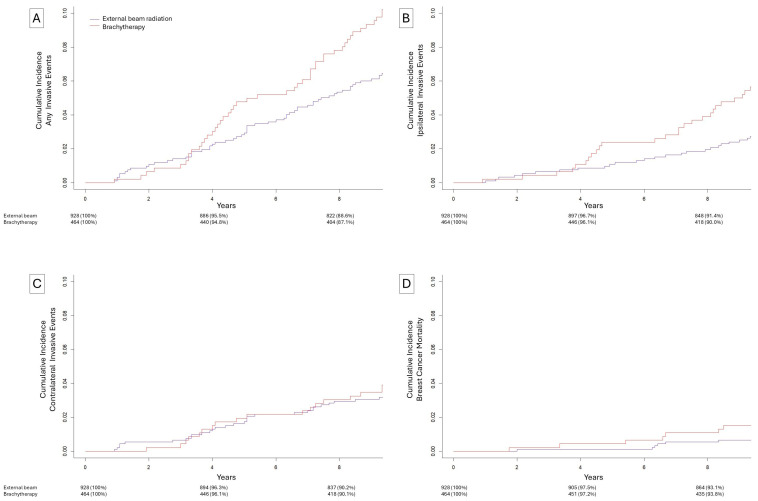
Cumulative incidence of events after external-beam radiation (blue) or brachytherapy (red). (**A**) Ipsilateral, contralateral, or breast cancer mortality subsequent events, (**B**) ipsilateral invasive subsequent events, (**C**) contralateral invasive subsequent events, (**D**) breast cancer specific mortality.

**Table 1 biomedicines-13-02823-t001:** Cohort characterization of matched patients undergoing breast-conserving surgery only, breast-conserving surgery with brachytherapy, or breast-conserving surgery with external-beam radiation.

	Breast-Conserving Surgery with Brachytherapy (N = 464)	Breast-Conserving Surgery Only (N = 928)	Breast-Conserving Surgery with External-Beam Radiation (N = 928)
Age			
Median (IQR)	60 (53–66)	60 (53–67)	60 (53–67)
Race			
White	386 (83.2%)	767 (82.7%)	771 (83.1%)
Black	43 (9.3%)	93 (10.0%)	93 (10.0%)
Other	35 (7.5%)	68 (7.3%)	64 (6.9%)
Grade *			
Low	46 (9.9%)	92 (9.9%)	92 (9.9%)
Intermediate	208 (44.8%)	416 (44.8%)	416 (44.8%)
High	161 (34.7%)	322 (34.7%)	322 (34.7%)
Undifferentiated	49 (10.6%)	98 (10.6%)	98 (10.6%)
Hormone receptor status *			
Positive	368 (79.3%)	736 (79.3%)	736 (79.3%)
Negative	96 (20.7%)	192 (20.7%)	192 (20.7%)
Size			
Median (IQR)	8 (4.75–14)	7 (4–14)	8 (4–15)
Adjuvant Radiation			
None	0 (0%)	928 (100%)	0 (0%)
Brachytherapy	464 (100%)	0 (0%)	0 (0%)
External-beam	0 (0%)	0 (0%)	928 (100%)
Adjuvant Endocrine			
Received	0 (0%)	0 (0%)	0 (0%)
Not received	464 (100%)	928 (100%)	928 (100%)

IQR = interquartile range, * Exact matching performed for these variables.

**Table 2 biomedicines-13-02823-t002:** Adjusted matched competing risk regressions for breast-conserving surgery only compared to breast-conserving surgery with brachytherapy.

	Any Invasive Event		Ipsilateral Invasive Event		Contralateral Invasive Event	
	sHR	*p*	sHR	*p*	sHR	*p*
Age						
≤55 years	Ref	-	Ref	-	Ref	-
>55 years	1.27 (1.52–3.72)	0.24	0.97 (0.60–1.57)	0.90	2.01 (0.97–4.16)	0.06
Race						
Non-Black	Ref	-	Ref	-	Ref	-
Black	2.37 (1.52–3.72)	<0.001	2.71 (1.56–4.70)	<0.001	1.00 (0.34–2.84)	1.00
Grade						
Low/Int	Ref	-	Ref	-	Ref	-
High/Undiff	1.07 (0.74–1.54)	0.73	1.06 (0.66–1.72)	0.81	0.75 (0.41–1.37)	0.35
Size						
≤10 mm	Ref	-	Ref	-	Ref	-
>10 mm	0.95 (0.66–1.37)	0.79	0.95 (0.59–1.51)	0.81	0.75 (0.40–1.43)	0.39
HR Status						
HR Positive	Ref	-	Ref	-	Ref	-
HR Negative	1.09 (0.70–1.70)	0.70	1.34 (0.78–2.31)	0.29	1.00 (0.47–2.14)	1.00
Treatment						
BCS	Ref	-	Ref	-	Ref	-
BCS + Brachy 0–3 years	0.28 (0.10–0.81)	0.02	0.19 (0.04–0.81)	0.03	0.29 (0.04–2.34)	0.24
BCS + Brachy 3–10 years	1.82 (1.20–2.76)	0.005	1.66 (0.96–2.86)	0.07	1.92 (0.53–0.98)	0.06

sHR = Subdistribution Hazard Ratio, Ref = Reference, Int = Intermediate, Undiff = Undifferentiated, HR = Hormone receptor.

**Table 3 biomedicines-13-02823-t003:** Adjusted matched competing risk regressions for breast-conserving surgery with brachytherapy compared to breast-conserving surgery with external-beam radiation.

	Any Invasive Event		Ipsilateral Invasive Event		Contralateral Invasive Event	
	sHR	*p*	sHR	*p*	sHR	*p*
Age						
≤55 years	Ref	-	Ref	-	Ref	-
>55 years	1.00 (0.67–1.47)	0.98	0.72 (0.41–1.24)	0.23	1.11 (0.62–1.99)	0.73
Race						
Non-Black	Ref	-	Ref	-	Ref	-
Black	2.44 (1.54–3.88)	<0.001	2.78 (1.47–5.24)	0.002	1.55 (0.70–3.43)	0.28
Grade						
Low/Int	Ref	-	Ref	-	Ref	-
High/Undiff	1.30 (0.88–1.92)	0.19	1.65 (0.91–2.99)	0.10	0.81 (0.45–1.45)	0.48
Size						
≤10 mm	Ref	-	Ref	-	Ref	-
>10 mm	0.70 (0.46–1.06)	0.09	0.60 (0.32–1.11)	0.10	0.88 (0.48–1.59)	0.67
HR Status						
HR Positive	Ref	-	Ref	-	Ref	-
HR Negative	1.19 (0.75–1.88)	0.46	1.65 (0.87–3.14)	0.12	0.83 (0.39–1.78)	0.63
Treatment						
BCS + External	Ref	-	Ref	-	Ref	-
BCS + Brachy 0–3 years	0.61 (0.20–1.87)	0.39	0.65 (0.13–3.23)	0.60	0.33 (0.04–2.75)	0.31
BCS + Brachy 3–10 years	1.73 (1.15–2.59)	0.009	2.20 (1.22–3.95)	0.009	1.36 (0.73–2.51)	0.33

sHR = Subdistribution Hazard Ratio, Ref = Reference, Int = Intermediate, Undiff = Undifferentiated, HR = Hormone receptor.

## Data Availability

The data presented in this study are openly available in SEER (https://seer.cancer.gov/data/access.html, accessed on 4 April 2024).

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
