# Peer review of "Relative Effects of Brachytherapy and Beam Radiation for DCIS on Subsequent Invasive Events"

_biomedicines, 2025, doi:10.3390/biomedicines13112823_

Round 1
Reviewer 1 Report
Comments and Suggestions for Authors
The main question of this manuscript is a comparison between two radiotherapy techniques which can be used for treatment and prevention of malignant recurrence in a rare cancer.
The topic is a literature review and evaluate 4 trails for the mentioned goal. In my opinion this kind of reviews can help to choose better technique in daily practice of the readers. A very good classification of DCIS patients, with a very nice notice to the colleagues who are involve in treatment of these patients. Of course, as you mentioned yourselves, this kind of trials should be done more for more accurate results.
As stated above such compares can be useful in clinic, there are not many published data in this object at all. The conclusion is reasonable and the references are appropriate.
Author Response
Reviewer #1
Comments and Suggestions for Authors
The main question of this manuscript is a comparison between two radiotherapy techniques which can be used for treatment and prevention of malignant recurrence in a rare cancer.
The topic is a literature review and evaluate 4 trails for the mentioned goal. In my opinion this kind of reviews can help to choose better technique in daily practice of the readers. A very good classification of DCIS patients, with a very nice notice to the colleagues who are involve in treatment of these patients. Of course, as you mentioned yourselves, this kind of trials should be done more for more accurate results.
As stated above such compares can be useful in clinic, there are not many published data in this object at all. The conclusion is reasonable and the references are appropriate.
We are greatly appreciative of Reviewer #1’s review of our manuscript and encouraging feedback. We are grateful for their recognition of the utility of the manuscript.
Reviewer 2 Report
Comments and Suggestions for Authors
Dear Authors,
Thank you for the opportunity to review this interesting and clinically relevant analysis of brachytherapy versus external‐beam radiation (EBRT) or omission of radiotherapy for DCIS using SEER. The focus on invasive subsequent events is valuable and the use of propensity matching with competing‐risk methods is appropriate. Below are detailed, constructive suggestions to help strengthen the manuscript.
1) Introduction and Framing
- The background is strong, but the framing would benefit from explicitly contrasting your focus on invasive events (the outcomes that most influence prognosis and systemic treatment) with prior randomized APBI/WBI trials that largely reported composite local recurrence.
- Bring current clinical guidance into the narrative more clearly, noting that modern recommendations allow PBI in selected DCIS, which underscores why your DCIS‐specific, invasive‐event focus is timely.
2) Cohort Definition and Generalizability
- You excluded patients who received endocrine therapy to avoid confounding, which is reasonable methodologically, but it narrows clinical applicability. Please state this explicitly up front: your findings best generalize to DCIS patients treated without endocrine therapy.
- Excluding patients with known HER2 status needs clearer rationale. Because HER2 testing patterns vary by era and center, excluding “known HER2” may introduce selection by calendar time or disease features. Consider a sensitivity analysis that (a) restricts to years before routine HER2 testing, and/or (b) includes known-HER2 cases with HER2 as a covariate, to show robustness.
- If possible, report the distribution of diagnosis years by treatment group. This will help the reader infer the likelihood that EBRT was mostly WBI in earlier eras.
3) Matching and Covariate Balance
- The matching strategy is appropriate. Please report standardized mean differences before and after matching for each covariate and a Love plot in the supplement. This simple addition demonstrates balance and reassures readers that the groups are comparable.
- Briefly justify the caliper choice and the 1:2 ratio, and mention the number of matched sets and any discarded cases.
4) Endpoints and Competing Risks
- Clearly define which events are treated as competing risks for each endpoint and ensure this is consistent throughout. For ipsilateral invasive events, it appears any death is a competing risk, whereas for “any invasive event” non–breast-cancer death is treated as competing only for invasive events. Make these choices explicit and consistent.
- For transparency, provide the number of events and person‐years by arm and by time window (0–3 years and >3 years) in a supplementary table. Absolute numbers alongside subdistribution HRs make interpretation easier.
- Consider also reporting cause-specific hazard models (as a sensitivity to the Fine–Gray approach) and providing 95% CIs for all 5- and 10-year cumulative incidences.
5) Time-Varying Treatment Effect
- The split at 3 years is clinically plausible, but please justify how it was chosen (e.g., graphical inspection, non‐proportional hazards tests, or pre‐specification).
- Provide formal tests of non‐proportionality and, if feasible, show flexible time-varying effects (e.g., piecewise at alternative cut points, or using time-varying coefficients with splines) to confirm your qualitative conclusion (early equivalence/reduction, later increase).
6) Residual Confounding and Unmeasured Variables
- Acknowledge explicitly that margins, focality, diagnostic modality (screen vs. symptomatic), and boost use are unavailable in SEER yet are known to influence recurrence.
- If possible, include proxies: e.g., re-excision or early reoperation codes as crude indicators of margin status; facility type or region as proxies for brachytherapy availability; and calendar year strata to partially account for evolving techniques.
- Consider performing a quantitative bias/sensitivity analysis (e.g., an E-value) to show how strong an unmeasured confounder would need to be to explain away the late hazard increase with brachytherapy versus EBRT.
7) EBRT vs PBI Clarification
- Because SEER cannot distinguish EB-PBI from WBI, interpret “EBRT” as predominantly WBI, especially in earlier eras, and state this clearly in the abstract and conclusions.
- Add a stratified analysis by diagnosis period (e.g., pre- and post-2015/2017) to illustrate whether the brachytherapy vs EBRT difference persists in more contemporary cohorts.
8) Quadrant/Location Analysis
- The exploratory quadrant analysis is intriguing. Please:
- Clarify the exact SEER variables and coding hierarchy used to assign quadrants for initial and subsequent lesions.
- Provide denominators and event counts, and indicate how “borderline” or “NOS” sites were handled.
- Use exact tests and present confidence intervals; label this analysis explicitly as exploratory and hypothesis‐generating given small numbers and potential misclassification.
- Consider showing early vs late (≤3 vs >3 years) quadrant patterns for each arm in a small supplementary figure or table.
9) Results Presentation and Internal Consistency
- Correct internal inconsistencies: the abstract lists an early-phase p-value that appears to disagree with the table; ensure all p-values and sHRs match across abstract, text, tables, and figures.
- There are a few typographical issues in tables (e.g., confidence interval formatting and p-values with stray characters). A careful proofread will fix these.
- Alongside sHRs, report absolute risk differences at 10 years with 95% CIs. Consider number needed to treat/harm to help communicate clinical magnitude.
10) Figures and Tables
- Ensure cumulative incidence plots include confidence bands and numbers-at-risk tables. This improves readability and supports critical appraisal.
- Check that colors/patterns remain distinguishable in grayscale.
- Revise Table titles/footnotes for precision (e.g., remove any reference to endocrine therapy in a table title if no patients received it).
- Add a supplementary table with baseline characteristics pre-matching to show how matching changed the distributions.
11) Interpretation and Language
- Your main conclusion - that brachytherapy did not reduce invasive ipsilateral events versus no RT and was inferior to EBRT in the later time window—is supported by your analyses. However, temper statements of inferiority or “suboptimal” with the observational nature of the data and the inability to separate EB-PBI from WBI. Phrases like “these findings suggest…” or “are consistent with…” would be more appropriate.
- Discuss the unexpected pattern for contralateral invasive events (non‐significant trend toward increase after 3 years with brachytherapy). As contralateral events should not be influenced by ipsilateral local therapy, potential explanations include differences in baseline risk, surveillance or coding differences, or random variation.
12) Clinical Context
- Briefly note that SEER cannot address toxicity, cosmesis, patient-reported outcomes, or detailed dose/technique factors that influence treatment choice between PBI and WBI.
- Add a short paragraph on how these findings might influence patient counseling today (e.g., invasive-risk reduction prioritizes whole-breast coverage in many DCIS cases without endocrine therapy), while calling for prospective validation.
13) Reproducibility and Reporting
- State the exact SEER submission used, extraction date, and all variable definitions/codes.
- Consider sharing analysis code (even as pseudo-code) and providing a STROBE checklist in the supplement.
- Report median follow-up and interquartile range by matched group to frame the timing of events.
14) Minor Editorial Points
- Standardize age and size cut points rationale (e.g., 55 years, 10 mm): state whether these were pre-specified or based on distributional considerations.
- Ensure consistent terminology: “ipsilateral invasive event,” “contralateral invasive event,” and “any invasive event” with or without breast cancer–specific mortality should be defined once and used consistently thereafter.
Overall, the manuscript is clearly written in an appropriate scientific register, with a logical progression from background to methods, results, and interpretation. The prose is generally concise and the terminology is used correctly, which makes the central message easy to follow. With that said, a careful language pass would noticeably improve polish and readability. I recommend a light professional copyedit focused on several recurring issues: first, remove minor typographical slips and duplicated words (there is at least one “that that” in the Introduction). Second, ensure complete internal consistency of numerical reporting—some p-values and estimates in the Abstract do not match the corresponding tables; these should be reconciled so that Abstract, Results, and tables all agree exactly. Third, there are a few small glitches in numerical typography within tables (e.g., malformed confidence intervals or stray characters) and an instance where a table title/footnote implies endocrine therapy despite its exclusion from the cohort; adjusting these will prevent reader confusion.
Consistency in style will also help. Please standardize hyphenation and compound modifiers (e.g., “external-beam radiation,” “whole-breast irradiation,” “partial-breast irradiation,” “non-Black”) and ensure uniform capitalization for receptor status (ER/PR). Abbreviations should be defined at first use in the main text (e.g., PBI, WBI, EBRT, sHR) and then used consistently thereafter. For statistical notation, use a uniform convention throughout (italic p, spaces around “=”, e.g., p = 0.02; sHR = 2.19), keep a space between numbers and units (e.g., 10 mm), and no space before the percent sign. Finally, consider splitting a few longer sentences—especially in the Introduction and Discussion where multiple clauses summarize prior trials or several limitations at once—so that key points stand out more clearly.
Author Response
Reviewer #2
Dear Authors,
Thank you for the opportunity to review this interesting and clinically relevant analysis of brachytherapy versus external‐beam radiation (EBRT) or omission of radiotherapy for DCIS using SEER. The focus on invasive subsequent events is valuable and the use of propensity matching with competing‐risk methods is appropriate. Below are detailed, constructive suggestions to help strengthen the manuscript.
We are very grateful for Reviewer #2’s thoughtful review or our manuscript and helpful feedback and we feel that the changes we have made in response to them has led to the improvement in the quality of the manuscript.
1) Introduction and Framing
- The background is strong, but the framing would benefit from explicitly contrasting your focus on invasive events (the outcomes that most influence prognosis and systemic treatment) with prior randomized APBI/WBI trials that largely reported composite local recurrence.
This is an excellent point. We have added the following:
“In contrast to the trials on PBI, we focus exclusively on ipsilateral invasive events, the most important prognostic recurrence type for patients with DCIS.”
- Bring current clinical guidance into the narrative more clearly, noting that modern recommendations allow PBI in selected DCIS, which underscores why your DCIS‐specific, invasive‐event focus is timely.
This is also an excellent point. We have added the following:
“These trials helped inform modern ASTRO guidelines, which offer a strong recommendation for PBI in patients with DCIS who are 40 years or older, have low or intermediate grade disease, and have lesion size 2 cm or less.”
2) Cohort Definition and Generalizability
- You excluded patients who received endocrine therapy to avoid confounding, which is reasonable methodologically, but it narrows clinical applicability. Please state this explicitly up front: your findings best generalize to DCIS patients treated without endocrine therapy.
This is a great point. We have added the following:
“We restricted our analysis to patients who did not receive endocrine therapy to isolate the effects of radiotherapy type, so our findings best generalize to DCIS treated without endocrine therapy.”
- Excluding patients with known HER2 status needs clearer rationale. Because HER2 testing patterns vary by era and center, excluding “known HER2” may introduce selection by calendar time or disease features. Consider a sensitivity analysis that (a) restricts to years before routine HER2 testing, and/or (b) includes known-HER2 cases with HER2 as a covariate, to show robustness.
This is a good point. The rationale for excluding patients with HER2 status is that for DCIS, HER2 testing has never been recommended, and should have only been performed for patients who participated in NSABP-43. However, Reviewer #2 is right, there is still center-dependent variability. While we agree with Reviewer #2 that a sensitivity analysis would be completely reasonable, we are unable to perform this in the 5-day window. There are relatively few patients who were excluded for this reason: for example, in the brachytherapy to no radiation comparison, 452 out of 8,412 (5.4%) patients had known HER2 status (HER2 status reporting started in 2010 and our time frame for inclusion ended in 2011). To address this valid point, we added the following line:
“Bias may have been introduced by excluding known HER2 status, though only a minority of eligible patients (~5%) were excluded for this reason”
- If possible, report the distribution of diagnosis years by treatment group. This will help the reader infer the likelihood that EBRT was mostly WBI in earlier eras.
This is a great point. Since we required hormone status which started collection in 2007 and required 10 years of potential for follow-up leading to a cutoff of 2011, we have added the following:
“Patients were treated between 2007-2011.”
3) Matching and Covariate Balance
- The matching strategy is appropriate. Please report standardized mean differences before and after matching for each covariate and a Love plot in the supplement. This simple addition demonstrates balance and reassures readers that the groups are comparable.
This is a good point by Reviewer #2, however we are unfortunately unable to complete this within the 5-day window.
- Briefly justify the caliper choice and the 1:2 ratio, and mention the number of matched sets and any discarded cases.
This is a great point, we have added the following:
“The 1:2 ratio was selected due to the relatively small numbers of patients who received brachytherapy relative to those not receiving radiotherapy or those receiving external-beam radiation.”
4) Endpoints and Competing Risks
- Clearly define which events are treated as competing risks for each endpoint and ensure this is consistent throughout. For ipsilateral invasive events, it appears any death is a competing risk, whereas for “any invasive event” non–breast-cancer death is treated as competing only for invasive events. Make these choices explicit and consistent.
This choice was deliberate and is consistent throughout the manuscript. The purpose of the ipsilateral invasive analysis, similar to the contralateral invasive analysis, was to identify the effects of each radiotherapy approach (omission, brachytherapy/presumptive PBI, and external-beam/presumptive WBI) on these respective outcomes. The purpose of the “any invasive event” was to identify the effect on any type of invasive recurrence. Breast cancer mortality in the absence of an invasive recurrence after breast-conserving surgery is presumptive metastatic recurrence per the SEER Inquiry guidance to registrars. So the invasive subsequent event category is intended to be a catch-all for invasive subsequent events—metastatic recurrences in the absence of invasive ipsilateral or contralateral events cannot be properly coded as “ipsilateral” or “contralateral” but they can be properly coded as a form of “invasive” recurrence. That is the rationale for the manner in which competing events are treated.
- For transparency, provide the number of events and person‐years by arm and by time window (0–3 years and >3 years) in a supplementary table. Absolute numbers alongside subdistribution HRs make interpretation easier.
This is reasonable but unfortunately we cannot complete this in the 5-day window.
- Consider also reporting cause-specific hazard models (as a sensitivity to the Fine–Gray approach) and providing 95% CIs for all 5- and 10-year cumulative incidences.
Similar to the prior comment, this is reasonable but not something we can address in this 5-day window.
5) Time-Varying Treatment Effect
- The split at 3 years is clinically plausible, but please justify how it was chosen (e.g., graphical inspection, non‐proportional hazards tests, or pre‐specification).
This is an excellent point. It was selected by graphical inspection. We have added the following:
“This time was selected by graphical inspection”
- Provide formal tests of non‐proportionality and, if feasible, show flexible time-varying effects (e.g., piecewise at alternative cut points, or using time-varying coefficients with splines) to confirm your qualitative conclusion (early equivalence/reduction, later increase).
This is a reasonable consideration but not something we can complete in the 5-day window.
6) Residual Confounding and Unmeasured Variables
- Acknowledge explicitly that margins, focality, diagnostic modality (screen vs. symptomatic), and boost use are unavailable in SEER yet are known to influence recurrence.
This is an excellent point. We had mentioned in the limitations margins, focality, and diagnosic modality but had not acknowledged use of boost which we have now added:
“There are several known confounders that were not available for our analysis: margin status, focality of disease/growth distribution, use of a radiation boost, family history or any known germline pathogenic mutations, and whether lesions were diagnosed by screening or by palpation.”
- If possible, include proxies: e.g., re-excision or early reoperation codes as crude indicators of margin status; facility type or region as proxies for brachytherapy availability; and calendar year strata to partially account for evolving techniques.
This is an interesting idea, unfortunately the public SEER data files that we have access to do not offer re-excision codes, nor do we have access to facility type or any meaningful region data. In the final cohort calendar year was between 2007-2011 so we suspect that it is less useful as a proxy.
- Consider performing a quantitative bias/sensitivity analysis (e.g., an E-value) to show how strong an unmeasured confounder would need to be to explain away the late hazard increase with brachytherapy versus EBRT.
We agree with Reviewer #2 that this is a reasonable consideration but again we are unable to perform this in the 5-day window.
7) EBRT vs PBI Clarification
- Because SEER cannot distinguish EB-PBI from WBI, interpret “EBRT” as predominantly WBI, especially in earlier eras, and state this clearly in the abstract and conclusions.
This is an excellent point. We have added:
“External-beam radiation is assumed to be whole-breast radiation for the majority of patients in this cohort diagnosed from 2007-2011.”
And:
“It was of comparable efficacy to external-beam radiation, assumed to predominantly be WBI,…”
- Add a stratified analysis by diagnosis period (e.g., pre- and post-2015/2017) to illustrate whether the brachytherapy vs EBRT difference persists in more contemporary cohorts.
This is an excellent idea by Reviewer #2, however all patients were treated between 2007-2011 (we recognize in response to other comments by Reviewer #2 that we did not include this date year range in the initial draft which was an oversight) so we suspect that a stratified analysis by diagnosis period may be less insightful.
8) Quadrant/Location Analysis
- The exploratory quadrant analysis is intriguing. Please:
- Clarify the exact SEER variables and coding hierarchy used to assign quadrants for initial and subsequent lesions.
We appreciate Reviewer #2’s interest in this exploratory analysis. We included the SEER variables/hierarchy used:
“Quadrants were derived from ICD primary site codes were used to determine this.”
- Provide denominators and event counts, and indicate how “borderline” or “NOS” sites were handled.
We added the following to explain how “borderline” and “NOS” sites were handled:
“For recurrence quadrant analysis, only sites with a defined quadrant (upper/lower and inner/outer) were included for this portion of the analysis, and patients with borderline and “not otherwise specified” sites were not considered.”
Regarding denominators:
“Among patients meeting these criteria, 11 of 20 (55.0%) of patients who did not receive adjuvant radiation had lesions in the same quadrant, 4 of 14 (28.6%) of patients who received brachytherapy had lesions in the same quadrant, and 4/10 (40.0%) of patients who received external-beam radiation had lesions in the same quadrant.”
- Use exact tests and present confidence intervals; label this analysis explicitly as exploratory and hypothesis‐generating given small numbers and potential misclassification.
It is an excellent point that we should list this as an exploratory analysis, which he have added:
“We performed an exploratory analysis regarding whether patients with well-defined index in situ and ipsilateral invasive recurrence were located in the same or different quadrants.”
Regarding the exact tests and confidence intervals, we did not calculate these because we had to exclude patients who did not have a well-defined index or recurrent quadrant as described above (e.g. borderline, NOS lesions) so we did not feel that including confidence intervals would be as useful in this context. We also had some concern that performing a formal statistical analysis to compare the groups might mislead readers into thinking that this is anything but an exploratory comparison.
- Consider showing early vs late (≤3 vs >3 years) quadrant patterns for each arm in a small supplementary figure or table.
This is another good point, but as with others we unfortunately cannot complete this in the 5-day window.
9) Results Presentation and Internal Consistency
- Correct internal inconsistencies: the abstract lists an early-phase p-value that appears to disagree with the table; ensure all p-values and sHRs match across abstract, text, tables, and figures.
This is an excellent catch by Reviewer #2 which we appreciate and have corrected.
- There are a few typographical issues in tables (e.g., confidence interval formatting and p-values with stray characters). A careful proofread will fix these.
This is also an excellent catch by Reviewer #2 which we similarly appreciate and have corrected.
- Alongside sHRs, report absolute risk differences at 10 years with 95% CIs. Consider number needed to treat/harm to help communicate clinical magnitude.
Similar to other comments, while we recognize that this would assist readers with assessing clinical magnitude, we unfortunately do not have time given the 5-day window coupled with other revisions to make these additions. The absolute 10 year risks at 10 years are, however, reported in-text and can also be observed in the cumulative incidence curves.
10) Figures and Tables
- Ensure cumulative incidence plots include confidence bands and numbers-at-risk tables. This improves readability and supports critical appraisal.
This is a good point by Reviewer #2. We included numbers-at-risk tables at 4-year intervals.
- Check that colors/patterns remain distinguishable in grayscale.
We had prepared the images in color because the Instructions for Authors page for Biomedicines stated that color figures are encouraged and are no extra charge, however if the figures will be printed in grayscale we can instead transition to dotted and solid lines to ensure no loss of information.
- Revise Table titles/footnotes for precision (e.g., remove any reference to endocrine therapy in a table title if no patients received it).
This is an excellent catch by Reviewer #2 and we have removed this reference.
- Add a supplementary table with baseline characteristics pre-matching to show how matching changed the distributions.
Similar to with several other items, we can provide this but would need more than the 5-day window to prepare it. Features of the full, unmatched cohort are described in Section 3.1, but we recognize that Reviewer #2 is correct in that a full supplementary table would provide more context.
11) Interpretation and Language
- Your main conclusion - that brachytherapy did not reduce invasive ipsilateral events versus no RT and was inferior to EBRT in the later time window—is supported by your analyses. However, temper statements of inferiority or “suboptimal” with the observational nature of the data and the inability to separate EB-PBI from WBI. Phrases like “these findings suggest…” or “are consistent with…” would be more appropriate.
This is a fair point. We had used the word “may” to try to highlight the retrospective nature but replaced the word suboptimal with higher risk.
- Discuss the unexpected pattern for contralateral invasive events (non‐significant trend toward increase after 3 years with brachytherapy). As contralateral events should not be influenced by ipsilateral local therapy, potential explanations include differences in baseline risk, surveillance or coding differences, or random variation.
This is an astute observation. We suspect that the reason this trend is present in the brachytherapy vs. no radiation analysis and not the brachytherapy vs. WBI analysis relates to the use of radiation itself. A trend of increased contralateral invasive risk in the radiotherapy groups of borderline significance was also noted in the prospective trials SweDCIS and EORTC 10853, though was not present in NSABP B17. (UK/ANZ similarly did not note an increased contralateral risk but notably their analysis was only first events, not cumulative incidences, and so cannot be directly compared.) We also have a separate manuscript/analysis under review at another journal that relates to this topic and suggests a similar explanation. To address this, we have added the following:
“We noted a borderline increased risk of contralateral invasive events among patients who underwent brachytherapy relative to those who did not receive radiation therapy in the 3-10 year time frame. There are multiple possible explanations for this observation. This could suggest differences in baseline global risk for breast cancer, and suggestive of confounding by unobserved variables such as family history. However, it should be noted that of the three trials of radiotherapy for DCIS that reported on cumulative contralateral invasive cancer risks, SweDCIS identified a borderline significantly elevated risk of contralateral invasive cancers, EORTC 10853 found a significantly elevated risk of contralateral invasive cancers, and only NSABP B17 did not identify an increased risk of contralateral cancers. This suggests that the borderline elevated increased risk may relate to radiation itself, rather than to PBI specifically.”
12) Clinical Context
- Briefly note that SEER cannot address toxicity, cosmesis, patient-reported outcomes, or detailed dose/technique factors that influence treatment choice between PBI and WBI.
This is a great point. We have added the following:
“SEER does not contain data regarding toxicity, cosmesis, patient-reported outcomes, or detailed dose or technique factors that can influence the choice between PBI and WBI.”
- Add a short paragraph on how these findings might influence patient counseling today (e.g., invasive-risk reduction prioritizes whole-breast coverage in many DCIS cases without endocrine therapy), while calling for prospective validation.
Also a great point. We added the following:
“Prospective validation of our findings is indicated given these limitations. However, given the previously discussed gaps in knowledge from the existing clinical trials, our results suggest that for patients who will not receive endocrine therapy after breast-conserving surgery for DCIS, WBI may be preferable to PBI when the priority is the reduction of risk in invasive subsequent events. For example, while the magnitude of benefit in invasive risk reduction likely does not justify WBI over PBI for the majority of patients who presently meet ASTRO-specified criteria for PBI, patients with low clinicopathologic risk but high molecular risk may want to consider WBI, and similarly patients who are on the younger end of the age criteria or who have a long life expectancy may similarly want to consider WBI given our finding of increasing risk with longer follow-up time.”
13) Reproducibility and Reporting
- State the exact SEER submission used, extraction date, and all variable definitions/codes.
Our manuscript includes the submission (“Data from the Surveillance, Epidemiology and End Results (SEER) 17 registry grouping (November 2023 submission) were used” but we have added the following line regarding the extraction date:
“Data were extracted in April 2024.”
- Consider sharing analysis code (even as pseudo-code) and providing a STROBE checklist in the supplement.
This is a fair point. We could provide pseudo-code for the analysis portion and a STROBE checklist in the supplement but would, as mentioned with other points, require more than 5 days to complete this.
- Report median follow-up and interquartile range by matched group to frame the timing of events.
This is a good point. We could provide this however similar to other items this would require more than the 5-day window turnaround time. The life table under Figures 1 and 2 coupled with the actual cumulative incidence curves, while not perfect, do give a sense of framing for the timing of events by each outcome type.
14) Minor Editorial Points
- Standardize age and size cut points rationale (e.g., 55 years, 10 mm): state whether these were pre-specified or based on distributional considerations.
This is a great point. We added the following:
“Age and size cut points were selected based on distributional considerations.”
- Ensure consistent terminology: “ipsilateral invasive event,” “contralateral invasive event,” and “any invasive event” with or without breast cancer–specific mortality should be defined once and used consistently thereafter.
We appreciate this observation. We suspect the specified lack of clarity relates to how any invasive event includes the outcome of breast cancer mortality. We have added the following clarifying sentence:
“Breast cancer mortality was also considered as a separate outcome.”
Comments on the Quality of English Language
Overall, the manuscript is clearly written in an appropriate scientific register, with a logical progression from background to methods, results, and interpretation. The prose is generally concise and the terminology is used correctly, which makes the central message easy to follow. With that said, a careful language pass would noticeably improve polish and readability. I recommend a light professional copyedit focused on several recurring issues: first, remove minor typographical slips and duplicated words (there is at least one “that that” in the Introduction). Second, ensure complete internal consistency of numerical reporting—some p-values and estimates in the Abstract do not match the corresponding tables; these should be reconciled so that Abstract, Results, and tables all agree exactly. Third, there are a few small glitches in numerical typography within tables (e.g., malformed confidence intervals or stray characters) and an instance where a table title/footnote implies endocrine therapy despite its exclusion from the cohort; adjusting these will prevent reader confusion.
These are great points. We removed the double “that.” We appreciate Reviewer #2 catching places where the sHRs and p values were inconsistent, which we have corrected. We also appreciate Reviewer #2 noticing the numerical typography issues and the endocrine therapy mislabel which we have fixed.
Consistency in style will also help. Please standardize hyphenation and compound modifiers (e.g., “external-beam radiation,” “whole-breast irradiation,” “partial-breast irradiation,” “non-Black”) and ensure uniform capitalization for receptor status (ER/PR). Abbreviations should be defined at first use in the main text (e.g., PBI, WBI, EBRT, sHR) and then used consistently thereafter. For statistical notation, use a uniform convention throughout (italic p, spaces around “=”, e.g., p = 0.02; sHR = 2.19), keep a space between numbers and units (e.g., 10 mm), and no space before the percent sign. Finally, consider splitting a few longer sentences—especially in the Introduction and Discussion where multiple clauses summarize prior trials or several limitations at once—so that key points stand out more clearly.
We have made the suggested changes in style, including changing to “external-beam”, “whole-breast”, and “partial-breast”, maintained “non-Black”, added spaces between “=” signs, italicized p values, and added spaces between numbers and units.
We are grateful for Reviewer #2’s thorough assessment of our manuscript and appreciate the recommended revisions which we believe have improved the quality of the manuscript. We recognize that we have not been able to incorporate all of Reviewer #2’s suggestions, many of which are secondary to time constraints, but they are all excellent suggestions and we have addressed as many of them as we can in the time frame allotted.
Reviewer 3 Report
Comments and Suggestions for Authors
This manuscript attempts to provide a comprehensive retrospective study of treatment outcomes comparing external beam versus brachytherapy irradiation as an adjuvant therapy for DCIS breast cancer patients. The study finds that patients treated with brachytherapy generally received suboptimal radiation therapy to prevent the recurrence of ipsilateral invasive disease after three years.
In general, this manuscript is well written and complete with a few minor suggestions for improvement as indicated below. The list of references seemed reasonable. I did not detect any evidence of plagiarism and did not use AI for the purposes of reviewing this manuscript.
Specific comments:
Line 49: It would be helpful if the authors summarized how the results were significantly different in terms of recurrent disease. For example, which group of patients had better (or worse) treatment outcomes with brachytherapy PBI?
Line 84: What was the selected age cutoff? 55 as stated later in the manuscript? Or 65?
Line 135: 2.7% of patient receiving external radiation…what was the percentage of patients receiving partial versus whole breast irradiation?
Table 2: It is unclear what “Ref” means to the reader. Please specify the units for the numbers. You do want the reader to second guess what these mean.
Discussion Section: It is also important to note that there are many different brachytherapy applicators that have been used over time. For example, the first balloon applicator (Mammosite) was a single catheter that treated a uniform 1 cm margin around the surface of the balloon. The Contura was an improvement where five channels could be used to better modulate the brachytherapy dose distribution, but still less ideal for treating lesions close to the skin bridge and chestwall. The SAVI and interstitial brachytherapy applicators offered greater conformality to the tumor cavity, but still only treated to a 1 cm target margin. Future studies should include an analysis of the differences in brachytherapy applicator types and their dosimetry because the more complex applicators such as SAVI could provide better prophylactic treatment coverage of sub-microscopic disease.
Furthermore, if WBI demonstrate better treatment outcomes, then perhaps the 1 cm target volume around the applicator surface should be reconsidered in brachytherapy procedures.
I don’t believe this study included IORT irradiation with Xoft which has also been used at other centers. These procedures also falls in the category of brachytherapy at some institutions.
Author Response
Reviewer #3
This manuscript attempts to provide a comprehensive retrospective study of treatment outcomes comparing external beam versus brachytherapy irradiation as an adjuvant therapy for DCIS breast cancer patients. The study finds that patients treated with brachytherapy generally received suboptimal radiation therapy to prevent the recurrence of ipsilateral invasive disease after three years.
In general, this manuscript is well written and complete with a few minor suggestions for improvement as indicated below. The list of references seemed reasonable. I did not detect any evidence of plagiarism and did not use AI for the purposes of reviewing this manuscript.
We greatly appreciate Reviewer #3’s thoughtful review of our manuscript.
Specific comments:
Line 49: It would be helpful if the authors summarized how the results were significantly different in terms of recurrent disease. For example, which group of patients had better (or worse) treatment outcomes with brachytherapy PBI?
This is an excellent question by Reviewer #3. The only one of the four trials to report on a post-hoc subgroup analysis was NSABP B39. We added the following to describe these findings:
“Post-hoc analysis from NSABP B39 suggested that patients with lesion size 11-20 mm benefited more from WBI, and patients with positive hormone receptor status as well as premenopausal patients borderline benefited more from WBI.”
Line 84: What was the selected age cutoff? 55 as stated later in the manuscript? Or 65?
The selected age cutoff for inclusion study was 75, the age 55 used later in the study was not used as an inclusion criteria but rather to separate age at diagnosis into categorical groups since treating it as a numerical variable would be more difficult for readers to interpret.
Line 135: 2.7% of patient receiving external radiation…what was the percentage of patients receiving partial versus whole breast irradiation?
Unfortunately this represents one of the limitations of our study—SEER does not separate radiation into PBI and WBI. It offers brachytherapy as a category, however, as well as external-beam, and given that this treatment cohort was diagnosed between 2007-2011, an underlying assumption, which is a limitation, is that patients who received external-beam radiation were receiving WBI.
Table 2: It is unclear what “Ref” means to the reader. Please specify the units for the numbers. You do want the reader to second guess what these mean.
This is an excellent point by Reviewer #3. Ref is intended to represent the reference group, we added this to the legend underneath the tables for clarity.
Discussion Section: It is also important to note that there are many different brachytherapy applicators that have been used over time. For example, the first balloon applicator (Mammosite) was a single catheter that treated a uniform 1 cm margin around the surface of the balloon. The Contura was an improvement where five channels could be used to better modulate the brachytherapy dose distribution, but still less ideal for treating lesions close to the skin bridge and chestwall. The SAVI and interstitial brachytherapy applicators offered greater conformality to the tumor cavity, but still only treated to a 1 cm target margin. Future studies should include an analysis of the differences in brachytherapy applicator types and their dosimetry because the more complex applicators such as SAVI could provide better prophylactic treatment coverage of sub-microscopic disease.
Furthermore, if WBI demonstrate better treatment outcomes, then perhaps the 1 cm target volume around the applicator surface should be reconsidered in brachytherapy procedures.
I don’t believe this study included IORT irradiation with Xoft which has also been used at other centers. These procedures also falls in the category of brachytherapy at some institutions.
These are all excellent points and observations by Reviewer #3. We have added the following:
“There have been changes in brachytherapy applicators over time, with differences among applicators in margin and conformality.”
“Intraoperative radiotherapy was not included in this study, but is a form of brachytherapy that is utilized at some institutions.”
“Alternately, our results may suggest that the margin of brachytherapy may need to be increased for patients treated for DCIS relative to patients treated for early invasive breast cancer.”